# Mapping Dithiobenzoate-Mediated RAFT Polymerization Products via Online Microreactor/Mass Spectrometry Monitoring

**DOI:** 10.3390/polym10111228

**Published:** 2018-11-06

**Authors:** Joris J. Haven, Tanja Junkers

**Affiliations:** 1Polymer Reaction Design Group, School of Chemistry, Monash University, 19 Rainforest Walk, Clayton, VIC 3800, Australia; joris.haven@monash.edu; 2Institute for Materials Research, Hasselt University, Martelarenlaan 42, 3500 Hasselt, Belgium

**Keywords:** online monitoring, flow polymerizations, dithiobenzoate-mediated RAFT polymerization, mass spectrometry, cross-termination

## Abstract

2-cyano-2-propyl dithiobenzoates (CPDB)-mediated reversible addition-fragmentation chain transfer (RAFT) polymerization was monitored by online flow microreactor/mass spectrometry. This enabled the reactions to be followed in a time-resolved manner, closely resolving product patterns in the reaction mixtures at any point in time. RAFT polymerization was investigated for low RAFT to monomer ratios, enabling the monitoring of the early stages of a typical RAFT polymerization. The expected transition from pre- to the RAFT main equilibrium is observed. However, very high abundancies for cross-termination products were also identified, both in the pre- and main equilibrium stage. This is a somewhat surprising result as such products have always been expected, but to date have not been observed in the majority of studies. Product isolation and NMR analysis revealed that cross-termination occurs in the para position of the benzoate ring and becomes fully irreversible via re-aromatization of the ring in a H-shift reaction. The present data suggest a pronounced chain-length dependence of the cross-termination reaction, which would explain why the products are seen here, but not in other studies.

## 1. Introduction

The reversible addition-fragmentation chain transfer (RAFT) polymerization technique, established by the CSIRO team in 1998, is one of the most widely used reversible-deactivation radical polymerization (RDRP) methods to synthesize polymers of controlled architecture and well-defined molecular weight [1]. Compared to other RDRP techniques such as atom-transfer [2] and nitroxide mediated radical polymerization [3], the overall radical concentration in RAFT is, in principle, not reduced, allowing for fast polymerizations, ideally identical in rate to their free radical polymerization counterparts (only ideal when the effect of difference in average chain length on diffusion is not taken into account) [4]. Despite its many advantages, RAFT kinetics has not been fully understood, especially when dithiobenzoates (Z–C(=S)S–R, Z = phenyl) are used as RAFT agents. 

Shortly after the introduction of the RAFT concept, rate retardation was observed with dithiobenzoates as control agent, raising a question about the reason for this deceleration (Scheme 1) [5,6,7,8]. Radicals are neither formed nor consumed by the RAFT equilibria, thus as long as the interchange of radicals in the degenerative transfer equilibria is sufficiently fast, no overall rate effect should be observed on the rate of polymerization. In any way, any observable rate retardation phenomenon must be caused by a reduction of the actively propagating radical concentration in comparison with conventional free radical polymerization (without the use of a RAFT agent), carried out under otherwise identical conditions. The associated mechanisms for interpreting this concentration reduction have since been under debate. Two models were proposed: (*i*) intermediate radical termination (IRT) of species **3** or **7** (Scheme 1) by itself or other transient radical species (so-called self-termination or cross-termination model) and (*ii*) slow fragmentation (SF) of the intermediate RAFT radical (species **3** or **7**, Scheme 1) due to the delocalization of the radical functionality into the phenyl ring by which a large proportion of radicals would be in a dormant state (slow-fragmentation model). Both processes are associated with a loss of propagating radicals and thus cause rate retardation. Although both the IRT and the SF model generally fit experimental data equally well and allow for good prediction of the expected molecular weight distributions, they are fundamentally in conflict. The fragmentation rate coefficient for the intermediate RAFT radical in both models differs by up to six orders of magnitude in the RAFT equilibrium constant (the ratio of addition of propagating radicals to the control agent over the fragmentation of the intermediate RAFT species) [9,10]. Further, on one hand, the SF model would predict a significant intermediate radical concentration, which is, however, not observed. On the other hand, the IRT model would predict very significant amounts of cross-terminated polymer in the final product mixture, which likewise cannot be detected in regular polymerizations. First indications of intermediate radical termination in the early stages of a real polymerization system was reported by Geelen et al. [11]. Several refinements on both theories were made over the years; nevertheless, none of them solved the riddle completely [12,13]. Therefore, as a consequence, various combinations of both models were suggested, taking chain length effects also into account, which further complicate the situation [14,15]. Furthermore, a reaction of propagating radicals with the irreversible cross-termination product was proposed as the so-called “missing reaction step” to resolve the question why no such product is found after polymerizations (Scheme 1) [16]. Over the years, an interesting debate evolved, which, however, until now remains unresolved. The reader is referred to excellent reviews for details on this matter [17,18,19,20,21]. 

In this manuscript, a microreactor mass spectrometer coupling is employed to monitor 2-cyano-2-propyl dithiobenzoate (CPDB)-mediated RAFT polymerizations of *n*-butyl acrylate and 2-2′-azoisobutyronitrile (AIBN) as a thermal initiator. Microreactor technology (MRT) chip reactors feature significant advantages compared to batch reactions such as ideal heat dissipation (due to the highly isothermal conditions provided) and high operation stability, to name a few [22]. A combination of both—continuous flow processing and online monitoring—constitutes an ideal tool in any chemical synthesis optimization and screening as time resolved data is obtained practically in real time during the reaction. Especially for polymerizations, ESI-MS has become a major and indispensable characterization tool to determine specific product patterns and online monitoring of product patterns gives a significant advantage in kinetic and mechanistic studies over conventional sampling methods.

Continuous flow RAFT polymerizations were screened for different reaction conditions (reagent concentrations, microreactor temperature and residence time). Single unit monomer insertion (SUMI) and irreversible cross-termination (CT) products were monitored in the early stages of the RAFT polymerization (pre-equilibrium). Microreactor residence time sweeps from 1 to 20 min were performed, which allowed for continuous nonstop monitoring of real-time synthesis products. In addition, the irreversible CT product was isolated and analyzed via nuclear magnetic resonance (NMR) spectroscopy. In here, the emphasis is put on the monitoring and observation of dithiobenzoate-mediated RAFT polymerization products, which can potentially give more insight into the complex and not yet fully understood RAFT mechanism. It is neither the intention to propose a solution nor instigate ongoing discussion about the rate coefficients that govern the RAFT equilibria; hence, the addition and the fragmentation rate.

## 2. Materials and Methods

### 2.1. Materials

The monomer *n*-butyl acrylate (Acros organics, 99%, Geel, Belgium) was deinhibited over a column of activated basic alumina prior to use. 2,2′-azobisisobutyronitrile (Sigma-Aldrich, 98%, Overijse, Belgium) was recrystallized twice from methanol prior to use. 2-cyano-2-propyl dithiobenzoate (CPDB, Sigma-Aldrich, 97+%) and *n*-butyl acetate (Acros organics, 99+%,) were used as received. 

### 2.2. Online MRT/ESI-MS Measurements

The Microreactor/ESI-MS set-up was described in detail previously [23]. In short, reactions take place in a conventional microreactor chip. When the microreactor is operated under true synthesis conditions, a reaction mixture is obtained at the reactor outlet that is unsuitable for MS analysis due to a mismatch in sample concentration, solvent, absence of doping agents, and flow rate. These issues can, however, be conveniently overcome by a strong dilution of the reactor mixture that exits the flow reactor with suitable doped ESI solvent mixtures, followed by a flow T-splitter to meet the requirements of the ESI-MS nozzle. Dilution also serves thereby as an effective solvent change next to the decrease in sample concentration down to the micromolar range. One of the many advantages of such a setup is the high flexibility in terms of concentrations and reaction conditions that can be investigated. A wide concentration window in the microreactor can be accessed; higher flow rates of increased sample concentration can be dynamically compensated by adjusting the dilution factor. 

### 2.3. Characterization Methods

Nuclear magnetic resonance (NMR, Varian, Grenoble, France) spectra were acquired in deuterated solvent (CDCl_3_) on a 400 MHz instrument. NMR spectra were analyzed via MestReNova software (Mestrelab Research G.L., Santiago de Compostela, Spain). All chemical shifts are recorded in ppm (δ) and determined relative to the residual solvent absorption peaks. The multiplicities were explained using the following abbreviations: s for singlet, d for doublet, t for triplet and m for multiplet.

Electrospray ionization mass spectrometry (ESI-MS) was performed using an LTQ orbitrap velos pro mass spectrometer (ThermoFischer Scientific, Geel, Belgium) equipped with an atmospheric pressure ionization source operating in the nebulizer assisted electrospray mode. The instrument was calibrated in the *m*/*z* range 220–2000 using a standard solution containing caffeine, MRFA and Ultramark 1621. A constant spray voltage of 5 kV was used and nitrogen at a dimensionless sheath gas flow-rate of 7 was applied. The capillary voltage, the tube lense offset voltage and the capillary temperature were set to 25 V, 120 V and 275 °C, respectively. For manual measurements, a polymer solution with concentration of 10 µg mL^−1^ was injected. A mixture of THF and methanol (THF:MeOH = 3:2), all HPLC grade, was used as solvent. Spectra were analyzed in Thermo Xcalibur Qual Browser software (ThermoFisher Scientific).

Purification of crude reaction mixtures was performed via flash column chromatography performed on a Büchi sepacore system (Büchi, Hendrik-Ido-Ambacht, The Netherlands) equipped with GRACE Resolve normal-phase silica cartridges (48 g).

### 2.4. Synthesis Procedures

General procedure for the online monitoring of RAFT polymerizations: The monomer *n*-butyl acrylate, 2-cyano-2-propyl dithiobenzoate (CPDB), 2,2′-azobisisobutyronitrile (AIBN) and butyl acetate were added into a glass vial. The glass vial was sealed by a rubber septum. The solution was degassed for 15 min by N_2_ purging, and subsequently inserted into the glovebox. The glass vial was opened and the solution was transferred to two gastight 1 mL syringes (SGE, Trajan Scientific Europe Ltd, Crownhill, UK). The syringes were employed to the ESI-MS microreactor setup and online monitoring experiments were subsequently started (see Section 2.2).

Synthesis of cross-termination product (see Section 3.4): 0.25 mmol (0.041 g, 0.5 equiv.) of 2,2′-azobisisobutyronitrile (AIBN), 0.5 mmol (0.110 g, 1 equiv.) of 2-cyano-2-propyl dithiobenzoate (CPDB) RAFT agent and 1 mL of butyl acetate were added into a glass vial with a stirring bar inside. The glass vial was sealed by a rubber septum. The solution was degassed for 15 min by N_2_ purging, and subsequently inserted into the glovebox. The glass vial was placed in a copper heat-block of 90 °C. The mixture was reacted for 2 h and subsequently quenched by cooling the vial in liquid nitrogen and subjecting to ambient atmosphere. Subsequently the mixture was transferred into an aluminum pan to evaporate excess of solvent. The mixture was purified by column chromatography (hexane: dichloromethane 4:1 to 1:1 *v*/*v*) yielding 0.022 g (13%) of pure CT product as a red oil. ^1^H NMR (300 MHz, CDCl_3_, δ): 7.59–7.54 (m, 2H, aromatic), 7.51–7.46 (m, 2H, aromatic), 5.47 (s, 1H, SC*H*S), 1.74 (s, 6H, C(*CH*_3_)_2_CN), 1.72 (s, 6H, C(*CH*_3_)_2_CN), 1.49 (s, 6H, C(*CH*_3_)_2_CN). ESI-MS (*m*/*z*): 380.12 (M + Na^+^). 

## 3. Results and Discussion

### 3.1. Online Microreactor/Mass Spectrometry Monitoring

Online mass spectrometry analysis of continuous flow processes provides real-time data and thus allows for rapid kinetic screening and in consequence efficient optimization of polymerization reactions. As schematically shown in Figure 1, reagent mixtures were prepared in gastight syringes and injected in the microflow reactor. Screening of the reaction can be done in minimal time and by usage of only trace amounts of material due to the small reactor volume (19.5 µL). The microreactor residence time depends on the flow rate as preset by the employed syringe pumps. Reaction products that exit the microreactor are transferred from the flow microreactor to the electrospray ionization (ESI) nozzle, which is directly achieved under constant flow conditions that allow for nonstop monitoring of the polymerization reaction, without the requirement of a sampling method. In this way, an unmatched information density with respect to product compositions and reaction kinetics is directly accessed in very short time spans. This information can be used for the detailed investigation of reaction mechanisms, but also for the rapid optimization of synthesis procedures. Data was acquired in situ and feedback provided to optimize the reaction outcome. The setup and its use were previously described in detail [23,24].

A useful feature of the microreactor/mass spectrometry setup is the ability to perform time sweep experiments, as already applied in our previous work [23]. In a time-sweep experiment, microreactor residence times are varied quickly by a sudden change in the preset flow rates of the embedded syringe pumps, followed by monitoring the outlet of the reactor. Typically, a range of different microreactor residence times (fixed flow rates) are screened in an online monitoring experiment. Different parameters (temperature, concentration and flow rate) can be chosen to comprehensively screen reaction efficiency. This way, the on-line setup allows for continuous ‘nonstop’ analysis of the reaction mixture at any given set of reaction conditions and reactor residence times. However, it only becomes really interesting if these flow rates are varied during the course of a microreactor polymerization without interrupting the continuous mass spectrometry measurement. Thus, a polymerization can be screened within a predetermined microreactor residence time interval. Time sweeps enable the operator to move forward or go back in time (as increased flow rates correspond to shorter reaction times), a feature that cannot be achieved in any batch reaction monitoring setup (see Figure 2). Practically any instance in time of a reaction can be directly imaged by sweeping over a range of reactor residence times, giving access to continuous data sampling with respect to product structure and composition under synthesis conditions. As example, when operating the setup at a flow rate of 0.975 µL/min (20 min residence time), flow rate can then be adjusted to 19.5 µL/min (1 min residence time) to perform a time sweep experiment from 20 to 1 min residence time. If now applied to a polymerization reaction, residence times can be sweeped forward and backwards (molecular weights will increase and decrease) until optimal conditions are identified. Note that a full sweep experiment always only takes as long as the lower flow rate that was chosen. Hence in the present example, a full kinetic screening is carried out within 20 min.

### 3.2. Real Time Monitoring Dithiobenzoate-Mediated RAFT Polymerizations

Dithiobenzoate-mediated RAFT polymerization processes were targeted via online microreactor/mass spectrometry monitoring. As a first demonstration on how the online monitoring setup can be employed, a reversible addition-fragmentation chain transfer (RAFT) polymerization (Scheme 2) of *n*-butyl acrylate (*n*BA) with the RAFT agent 2-cyano-2-propyl dithiobenzoate (CPDB) and 2-2′-azoisobutyronitrile (AIBN) as thermal initiator was screened. 

To investigate the kinetics and species formed during the early stages of the RAFT polymerization, typically, samples are taken during the course of polymerization whereby endgroup patterns are checked offline via soft ionization mass spectrometry methods. Here, the polymerization was continuously monitored by the online setup for different conditions (temperature, residence time and reagent concentrations) and ESI-MS spectra were analyzed accordingly. Table 1 highlights 8 screening results for the CPDB RAFT polymerization reactions with 1 equivalent of the monomer *n*-butyl acrylate and 0.1 or 0.25 equivalents of the thermal initiator AIBN. Screening equimolar RAFT to monomer concentrations allows us to monitor species formed during the early stages of the RAFT polymerization, before the main equilibrium of the process is reached. Polymerizations were screened at a microreactor temperature of 80 and 100 °C and microreactor residence times of 5 and 10 min. Product yields (%) for the species identified in Table 1 were determined by ESI-MS peak intensities (relative abundances) according to the following equation:Yield **X** = [MS peak intensity **X/**Total MS peak intensity] × 100

All individual species are calculated relative to the sum of the ESI-MS intensities of all species observed, an evaluation method that has been used and validated for series of polymer product analysis before [25,26]. It has to be noted though that ESI-MS is a qualitative analysis technique due to mass and ionization biases and is therefore not representative for absolute concentrations of species in the reaction mixture. This can be overcome by a complex labor-intensive calibration of the mass spectrometer, as shown previously. However, in this manuscript, ESI-MS results are described based on the relative abundance of the species rather than absolute concentrations.

After screening equimolar conditions of CPDB RAFT agent and *n*BA monomer, single unit monomer insertion (SUMI) products and the CPDB RAFT agent are identified in the reaction mixture for all screening conditions (Table 1). Furthermore, irreversible termination products (so-called cross-termination, CT) of the intermediate (macro)RAFT agent with propagating radicals present in the reaction mixture were observed. In Table 1, single unit monomer insertion products are indicated as SUMI_x_ (x = 1, 2, …) according to the amount of monomers inserted respectively and cross-termination products as CT_x_ (x = 0, 1, …) with CT_0_ being the irreversibly terminated intermediate CPDB RAFT by AIBN radical fragments during the pre-equilibrium stage (species **5**, Scheme 1). Unexpected rate retardation is a well-known phenomenon in dithiobenzoate-mediated RAFT polymerizations as discussed before. Observing irreversible termination in the early stages of the process is a direct proof of this cause. Rate constants of the RAFT equilibrium are not accessible via online mass spectrometry monitoring; however, kinetic studies were published that support the slow fragmentation of the intermediate RAFT radical (species **3** or **7**, Scheme 1). Chernikova et al. used an electron spin resonance (ESR) spin-trapping technique for quantitative detection of radicals (to measure addition and fragmentation rate coefficients for the RAFT equilibrium) to obtain an equilibrium constant in the order of 10^8^ L mol^−1^ [27]. In another contribution, Ranieri et al. showed via a ESR spin-trapping methodology that the stable RAFT intermediate radicals are formed in the early stages of the RAFT polymerization when dithiobenzoates are employed as controlling agents as stipulated by the so-called slow fragmentation theory [28].

Scheme 3 shows the chemical structures of the expected species with their corresponding monoisotopic masses observed during MS screening as discussed in Table 1. CT_1_ represents termination of the intermediate (macro)RAFT species where one *n*-butyl acrylate monomer is inserted. However, mass spectrometry cannot distinguish where the monomer unit is built in as both possible structures are isobaric (see Scheme 3). CPDB RAFT agent, SUMI_1_, CT_0_ and CT_1_ are identified for all conditions screened in Table 1. Very high amounts of AIBN were used in the study to allow for observation of significant peak intensities in the ESI-MS. Overall, as expected for higher residence times and reactor temperature a higher intensity of SUMI_1_ species is observed and less CPDB RAFT agent is present is the mixture. A direct relation is observed between the amount of CT_0_ species formed and the number of AIBN radicals generated. CT products are directly related to the number of radicals generated by AIBN so more CT is observed in the reaction mixture if the AIBN concentration is increased from 0.1 to 0.25 equivalents. Since the decomposition rate of AIBN (half-life time) is directly related to the reaction temperature more CT products are formed at 100 °C compared to 80 °C, simply due to the increased radical concentration and therefore cross-termination, and termination events in general, are more likely to occur.

Figure 3 shows the ESI-MS spectra for condition **3** (0.1 equiv. AIBN) and **7** (0.25 equiv. AIBN) in Table 1 at 100 °C and 5 min residence time. For these conditions, a high amount of CT_0_ product is formed which again is related to the radical concentration in the reaction mixture. In the first 5 min of the polymerization (during first half-life time of AIBN, t_1/2_ = ~8 min), more radicals are generated and radicals are more likely to terminate. Over the course of the polymerization, less AIBN radicals are generated and the relative abundance of CT_0_ decreases over time. This effect is nicely illustrated in the relative abundance of SUMI_1_ and CT_0_ in Table 1. Furthermore, SUMI_2_ products are not observed at the conditions screened in Table 1; higher SUMI products are only expected once the CPDB RAFT is fully consumed (so-called inhibition period). In addition, CT_1_ species are only observed with increasing radical concentration (thus high initiator concentration and temperature) or upon increasing the residence time (condition **8**). It has to be noted that by referring to a decrease in CT_x_ products in the results reported it actually refers to an increase of other (mostly likely SUMI) species in the polymerization mixture, which relies on the assumption of irreversible cross-termination; however, the reaction of propagating radicals with the irreversible cross-termination product has been proposed in literature [16].

Table 2 shows results for screening the same dithiobenzoate-mediated polymerization reaction as shown in Scheme 2 with five times increased monomer concentration to identify higher SUMI and CT species. Here, the same behavior is observed and similar conclusions can be drawn for condition **9**, **10** and **11** compared to the conditions in Table 1 (as discussed earlier). The RAFT polymerization shows inhibition until all CPDB RAFT agent is consumed. However, at 100 °C for 20 min (condition **12**) a clear transition is observed from the pre -to the main equilibrium in the RAFT polymerization process compared to condition **11**. At this stage all RAFT agent is converted into (macro)RAFT and higher SUMI and CT products (SUMI_2_, SUMI_3,_ CT_1_ and CT_2_) are observed in high amounts. Thus, cross-termination does not only occur in the pre-equilibrium stage. However, reactions are only screened for lower chain lengths, which may explain why in previous studies these species could not be identified. At chain lengths above 10 these might not occur in significant abundancies anymore, as severe chain length effects of the RAFT equilibrium constant have indeed been proposed both from ab-initio calculations, but also based on kinetic modelling [14,15,29]. 

### 3.3. Time Sweeping the Early Stages of Dithiobenzoate-Mediated Polymerizations

After identifying polymerization species formed during the early stages of dithiobenzoate-mediated polymerization, time sweep experiments were performed (see Section 3.1) to investigate the change in CT product formation in time (and chain length). Here, the polymerization was continuously monitored by ESI-MS while sweeping over a given residence time range. Solutions of CPDB RAFT agent, nBA monomer and AIBN in butyl acetate (BuOAc) were prepared in gastight syringes and employed in the microreactor/MS setup. Initial conditions applied to the microreactor were set to 1 min microreactor residence time (flow rate = 19.5 µL/min) and at a reactor temperature of 100 °C. The polymerization reaction is then recorded by increasing the residence time in the microreactor from 1 to 20 min (flow rate from 19.5 to 0.975 µL/min). Spectra were continuously acquired for the next 20 min and a continuous set of kinetic data is obtained based on the specific product patterns recorded for each individual residence time (MS spectra acquired every 2.2 s). Note that the dead volume between the microreactor exit and ESI-MS nozzle is not included (in practice data acquisition happens for 20 min + dead time of ~6 min). Such intense screening is not feasible with any other offline sampling method since the polymer sample needs to be quenched at the exact aimed residence time, purified and manually measured by ESI-MS, assuming all products are stable. Care has to be taken when performing these measurements, as the amount of data acquired in a short time interval (20 min) for each polymerization is enormous. As an example to illustrate this further, in this manuscript an ESI-MS spectrum from the polymer mixture that exits the microreactor was measured every 2.2 s. In total, 545 ESI-MS spectra are acquired in a time sweep from 1 to 20 min. To obtain information on a single species present in the spectra, peak intensities for all species present have to be extracted. 

Via this technique, the formation of cross-termination products over time for a range of reaction conditions (varying reagent concentrations and reactor temperature) is easily followed. In Figure 4 time sweep experiments are shown for different *n*BA:CPDB:AIBN ratios. Furthermore, the absolute concentration of the CPDB RAFT agent was varied and a clear effect on the reaction kinetics was observed. The difference in reagent concentrations can be explained by the microreactor setup whether the reaction mixture is prepared in one gastight syringe or, in case of screening different equivalents of monomer, two gastight syringes were applied. Overall, a similar trend is observed compared to the conditions screened in Table 1 and Table 2. CT products increase with increasing radical concentration in the reaction mixture, thus with increasing initiator concentration, higher SUMI and CT products are only formed when the CPDB RAFT agent is almost fully consumed (Figure 4D). In Figure 4A,C,E,F, the initial RAFT agent concentration is 0.25 M, therefore, by varying the [*n*BA] and [AIBN] concentration, a clear effect on the reaction rates of SUMI_x_ and CT_x_ could be observed. All reactions were performed at 100 °C in butyl acetate. In Figure 4C, a 5-times higher monomer concentration was screened compared to Figure 4A (initiator and RAFT concentration was kept constant). As expected, the formation of the SUMI_1_ product happens faster in 4C; however, almost no CT products were observed, while 4A shows the formation of CT_0_ in significant amounts from the start. The same effect is observed in Figure 4E,F for a lower overall initiator concentration. In Figure 4B,D, a higher initial RAFT agent concentration is used, which clearly influences the kinetic behavior in the pre- and main equilibrium. The higher RAFT agent concentration in Figure 4B causes the formation of CT_2_ compared to no CT products in Figure 4C. This can be explained by the higher initial concentration of the AIBN initiator (0.0625 M in Figure 4C compared to 0.1563 M in Figure 4B); cross-termination is therefore much more favored. In Figure 4B,D, the RAFT agent is fully consumed after approximately 12 min residence time and higher SUMI products were observed. 

### 3.4. Proposed Chemical Structure of the Cross-Termination Product

RAFT polymerizations mediated by dithiobenzoates experience rate retardation, mainly due to resonance stabilization of the intermediate radical (**3** and **7**, Scheme 1). Delocalization of the radical functionality affords resonance structures with reduced steric hindrance for the formation of cross-termination products by radical-radical coupling, as shown in Scheme 4. Previous discussions in literature focused on the cross-termination between propagating radicals and the delocalized radical (Scheme 4 to end up with a mixture of several regioisomers [13]. Observation of these products demonstrates that CT products are stable, and not prone to transfer reactions, where a CT product is transformed back into a regular RAFT polymer (see the “missing reaction step”) [16,30]. If such transfer is not occurring at low chain lengths, it is not evident why it should occur at higher degrees of polymerization. Observation of CT products in this study for early stage RAFT polymerization would thus underpin that the kinetic situation at the early stage of the main equilibrium is significantly different from later stages and that pronounced chain length effects do exist. Since termination is diffusion controlled, it is straightforward to assume that cross-termination will become less significant with increasing radical lengths. However, the equilibrium constant might also be prone to change. This is an effect that the current monitoring technique is, however, not able to catch. 

In a last step, we aimed at confirming the exact structure of the CT product, as ESI-MS does not allow to discern between several possible isobaric product structures (structural isomers). To reach this aim, a simple and straightforward batch experiment was designed. CPDB RAFT agent was heated in the presence of high amounts of AIBN (0.5 equivalents) dissolved in butyl acetate. The high concentration of AIBN radicals generated upon heating the mixture for 2 h at 90 °C will undergo radical termination events by radical combination, while still also taking part in the degenerative transfer equilibrium with the RAFT agent. In such a scenario, intermediate radical species (**3** and **7**, Scheme 1) are constantly formed upon addition of radical fragments, and concomitantly very likely to be terminated with further cyanoisopropyl fragments. Although this experiment is not performed under real polymerization conditions, it gives good insight into the processes that cause a loss of radicals, and in consequence rate retardation, during dithiobenzoate-mediated polymerization. After the reaction, the product mixture was purified by flash column chromatography and the cross-termination product (Figure 5B) and CPDB RAFT agent (Figure 5A) were isolated. Figure 5 shows NMR and ESI-MS analysis of the crude and purified reaction mixture. Details about the synthesis is provided in Section 2. Surprisingly, the main CT product (over 95%) observed is the irreversible radical-radical coupling on the para-position (Species **5** and **8**, Scheme 1), followed by a re-aromatization (H-shift) to regain its aromaticity (Species **6** and **9**, Scheme 1). Thus, it can be safely concluded that radical-radical coupling occurs exclusively on the para position as expected but re-aromatization via an H-shift occurs, which is not fully unexpected since it is much more energetically stable. HETCOR NMR (Figure 5C) analysis confirmed the re-aromatization by coupling between the ^13^C-^1^H of the dithioester moiety. No other coupling patterns were observed in heteronuclear correlation (HETCOR) NMR spectroscopy for this particular proton shift. The isolated re-aromatized para terminated cross-coupling product has a purity of +95% according to ^1^H NMR analysis. Termination in ortho position can be excluded, since no corresponding coupling was observed in NMR analysis (^1^H + HETCOR NMR). Figure 5D,E show MS of the crude and isolated re-aromatized cross-termination product, respectively.

## 4. Conclusions

The early stages of CPDB RAFT polymerization of an acrylate was monitored closely via online ESI-MS spectrometry coupled to a microreactor. In this way, time resolved data could be obtained for the reactions occurring. The so-obtained data show very clearly that next to propagation of the oligoacrylate chains, very significant amounts of so-called cross-termination events also occur. This is a remarkable result, as such species are typically not observed during RAFT polymerization, despite the strong evidence from reaction modelling that this reaction must occur. It may be believed that the reason for this mismatch is found in a profound chain-length dependence of the reaction. In the present study, very short chain lengths have been studied in contrast to somewhat longer chain lengths under investigation in previous studies. Time-resolved data shows clearly the rise of cross-termination products with time and progressive decay of the thermal initiator in the system. At least for the small chain lengths under observation here, it may thus be concluded that a transfer step as proposed before to explain the absence of cross-termination products is not operational nor needed to explain the data. Further, the structure of the cross-termination product was elucidated by product isolation and in-depth NMR analysis. Termination occurs on the para position of the benzoate and is followed by re-aromatization of the CT product in an H-shift reaction, forming a stable product. It should be noted that in this study no conclusions can be drawn about the magnitude of the RAFT equilibrium constant. Nevertheless, with the semi-quantitative data at hand from this study, it can be hoped that future modelling efforts will allow for a more refined picture, and consequently also for a better estimation of the true equilibrium constant. In any way, the present study gives strong evidence for a very pronounced chain length dependence over the first few propagation steps, confirming previous models by the Perrier group [15].

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
