# Peer review of "Mapping Dithiobenzoate-Mediated RAFT Polymerization Products via Online Microreactor/Mass Spectrometry Monitoring"

_polymers, 2018, doi:10.3390/polym10111228_

Round 1

Reviewer 1 Report

The manuscript deals with detailed mechanistic investigations into dithiobenzoate mediated RAFT polymerizations of RAFT. The information are obtained by inline monitoring of the reactions using a microreactor/MS set-up. The set-up allows for very fast screening of suitable reaction conditions. The results may contribute to resolving a long-standing discussion on the retardation of dithiobenzoate mediated RAFT polymerizations. The topic is of high importance and is certainly suited for publicaton in Polymers.

The excellent manuscript gives a good overview on the relevant literature and provides a detailed discussion of the experiments and the results obtained. I suggest to publish the manuscript after a few small changes.

Line 29: non-arguably instead of arguably?

Line 40: I think “if” should be replaced by “is”.

Line 106: The text reads “…by a strong dilution of the reactor flow mixture…” It is not clear, whether the original reaction mixture is diluted, prior to polymerization?

Line 372: The text reads “… to discern between several possible isobaric product structures.” The word isobaric in this sentence is not clear to me.

Author Response

We are thankful for the comments given. We have corrected the indicated typos and rephrased the respective sentences for better clarity in order to avoid misunderstandings.

Reviewer 2 Report

Haven and Junkers present an interesting study, highlighting that microreactor technology with detailed analysis allows a unique polymer product labeling. Moreover, new mechanistic insights are highlighted. I have only the following minor comments:

L 34; aspect of rate of RAFT polymerization vs. FRP. This is in an ideal context. There is of course the aspect of a difference of average chain length so that diffusional limitation on termination that are unavoidable have a different impact. In RAFT polymerization the chain lengths are lower and thus there is a less pronounced gel-effect possible. Perhaps this aspect can also be included (see e.g. De Rybel et al. Chem. Eng. Sci. 2018, 177,  163)

L 68 the recent update in Macromol. Theory Simul. 2017, 26, 1600048 could be also highlighted. Also in a general RAFT context: Polymers 2018, 10(3), 318.

L 73 I suggest to cite Macromol. Rapid Commun. 2015, 36, 2149 in view of the excellent temperature control

L 90 Perhaps good to mention the typical ranges as reported by several groups (e.g. Barner-Kowollik Junkers, Zhu, …)

Introduction: perhaps for a general reader Scheme 1 needs some more background as the typical star by recombination is not highlighted.

L 409 what about the conditions in view of reaction probabilities so regardless of chain length dependence

L 423 reference needed

Author Response

For reviewer 2 a point-by-point reply is given below:

"Haven and Junkers present an interesting study, highlighting that microreactor technology with detailed analysis allows a unique polymer product labeling. Moreover, new mechanistic insights are highlighted. I have only the following minor comments:"

We are grateful for the encouraging comment.

L 34; aspect of rate of RAFT polymerization vs. FRP. This is in an ideal context. There is of course the aspect of a difference of average chain length so that diffusional limitation on termination that are unavoidable have a different impact. In RAFT polymerization the chain lengths are lower and thus there is a less pronounced gel-effect possible. Perhaps this aspect can also be included (see e.g. De Rybel et al. Chem. Eng. Sci. 2018, 177,  163)

We have added a note on chain length and diffusion aspects and added the respective reference. 

L 68 the recent update in Macromol. Theory Simul. 2017, 26, 1600048 could be also highlighted. Also in a general RAFT context: Polymers 2018, 10(3), 318.

The given references have been added.

L 73 I suggest to cite Macromol. Rapid Commun. 2015, 36, 2149 in view of the excellent temperature control

The given reference has been added.

L 90 Perhaps good to mention the typical ranges as reported by several groups (e.g. Barner-Kowollik Junkers, Zhu, …)

We slightly rephrased the sentence for better clarity. We wish, however, not to give numbers here as we do not wish to instigate a discussion on these numbers in our paper - and by choosing references and giving numbers we would need to do so. We have given references to several reviews on the topic so the reader has access to a general overview on the topic.

Introduction: perhaps for a general reader Scheme 1 needs some more background as the typical star by recombination is not highlighted.

We believe that the scheme is clear. The star formation by recombination is indeed given (species 8).

L 409 what about the conditions in view of reaction probabilities so regardless of chain length dependence

We are unsure what the reviewer wishes to express here. We report on a factual change in the product composition with chain length - how can this not be taken into account? This is unclear to us, hence we made no change.

L 423 reference needed

A reference has been added as requested.